# Design of stable and self-regulated microbial consortia for chemical synthesis

Xianglai Li [1], Zhao Zhou [1], Wenna Li [1], Yajun Yan [2], Xiaolin Shen [1], Jia Wang [1], Xinxiao Sun [1✉] & Qipeng Yuan [1✉]

Microbial coculture engineering has emerged as a promising strategy for biomanufacturing. Stability and self-regulation pose a significant challenge for the generation of intrinsically robust cocultures for large-scale applications. Here, we introduce the use of multi-metabolite cross-feeding (MMCF) to establish a close correlation between the strains and the design rules for selecting the appropriate metabolic branches. This leads to an intrinicially stable two-strain coculture where the population composition and the product titer are insensitive to the initial inoculation ratios. With an intermediate-responsive biosensor, the population of the microbial coculture is autonomously balanced to minimize intermediate accumulation. This static-dynamic strategy is extendable to three-strain cocultures, as demonstrated with de novo biosynthesis of silybin/isosilybin. This strategy is generally applicable, paving the way to the industrial application of microbial cocultures.

[1] State Key Laboratory of Chemical Resource Engineering, Beijing University of Chemical Technology, Beijing 100029, China. [2] School of Chemical, Materials and Biomedical Engineering, College of Engineering, The University of Georgia, Athens, GA 30602, USA. ✉email: sunxx@mail.buct.edu.cn; yuanqp@mail.buct.edu.cn

Microbial production of industrial chemicals from renewable resources is an important direction for a sustainable economy[1]. Engineered single microbes have had significant success in the production of a variety of chemicals[2–5]. However, the metabolic capacity of single strains is limited, particularly for the production of chemicals with long biosynthetic pathways and complex structures. Modular coculture engineering has emerged as a promising approach to expand the capacity of microbial synthesis, having the distinct advantages of alleviating metabolic burden by division of labor, optimizing enzyme expression and catalysis by compartmentalization, simplifying the pathway optimization with modularization, and ability to utilize complex substrates[6–13].

Microbial consortia with stable and tunable population compositions are highly desirable for efficient production and process scale-up. Although developing microbial cocultures has recently attracted considerable interest, it is still challenging to rationally engineer a stable microbial consortium. To improve the stability, strategies applied include (1) decreasing the competition among the strains by using separate carbon sources[14–18] and (2) enhancing the intercellular correlation by establishing symbiotic relationships[15,19–23]. For example, two E. coli strains were engineered to utilize glucose and xylose, that cross-fed each other with tyrosine and phenylalanine for salidroside production[10]. In another example, an E. coli-Saccharomyces cerevisiae coculture was produced for the biosynthesis of oxygenated taxanes, in which S. cerevisiae uses acetate produced by E. coli for growth and, in turn, releases its inhibition on E. coli[24]. However, the coculture systems designed to date require adjustment of the initial inoculation ratios (IIRs) to optimize, severely limiting use for large-scale production[25,26].

Previous symbiotic relationships have been established primarily based on single-metabolite cross-feeding[15,19–23]. However, the secreted amount of a single metabolite is often not sufficient to support normal growth, limiting the robustness and stability of the coculture[27]. Self-regulation is another desired feature of a robust microbial coculture, where the strain ratio is autonomously coordinated to increase metabolic flux to the product with a minimal build-up of the intermediates. Self-regulation can be achieved by establishing dynamic population control systems using genetic circuits. Quorum sensing-based synthetic devices, such as cell lysis circuits and metabolic toggle switches, have been applied to modulate the population and metabolic activity of cocultures[28–32]. This, though, is pathway-independent and unable to respond to metabolic intermediates. In addition, the complicated genetic circuits may increase the metabolic burden, hampering their practical application. Population control can be realized with metabolite-responsive biosensors (MRBs). MRBs, including transcriptional- and ribo-switch-based MRBs, have been used for dynamic control of gene expression in monocultures[33,34]. However, their implementation in coculture engineering has not been reported.

In this study, we introduce the concept of a multi-metabolite cross-feeding (MMCF) strategy to strengthen the correlation between the microbial entities. Central to this strategy is choosing the proper metabolic branches for cross-feeding involving multiple metabolites (diversity) essential to cell growth (essentiality) and transferable across the cell membranes (transferability). Accordingly, we select the amino acid anabolism and the energy metabolism to establish a close cell-cell correlation (Fig. 1), resulting in a very stable coculture system. The lack of biosensors with appropriate specificity and dynamic range is a major obstacle to obstacle to achieve self-regulation of the cocultures. We characterize a caffeate-responsive biosensor, integrate it into the coculture system, and achieve autonomous regulation of strain ratios, yielding a significant increase in coniferol

production. Further, we apply this static-dynamic strategy to a three-strain coculture system and demonstrate de novo biosynthesis of silybin/isosilybin. This work provides a generalizable strategy for constructing robust coculture systems.

## Results

**MMCF enables a highly stable microbial coculture.** We first aimed to establish a stable coculture system whose population composition is insensitive to the IIRs and can converge onto a narrow range, which would greatly reduce the production fluctuation and facilitate scale-up. We used the prokaryotic model organism E. coli to construct a homogenous coculture and stepwise improved the stability by enhancing the correlation between the strains (Fig. 1). E. coli BW25113 and BW25113$\Delta pykA\Delta pykF$ were used as the basal strains. To reduce competition, two derivatives were generated to use glycerol and glucose, respectively. In E. coli, glucose is converted to glucose-6-phosphate via the phosphotransferase system (PTS) or glucokinase[35]. To block glucose catabolism, the relevant genes ptsG (encoding the glucose-specific transporter of the PTS), manXYZ (encoding the mannose PTS that can also transport glucose), and glk (encoding glucokinase) in BW25113$\Delta pykA\Delta pykF$ were deleted, generating the glycerol-utilizing strain Bgly1. Noteworthy, with the deletion of the two pyruvate kinase genes (pykA and pykF), the carbon flux would enter the TCA cycle primarily via the phosphoenolpyruvate carboxylase (encoded by ppc) by-pass. To block glycerol catabolism, gene glpK (encoding glycerol kinase) in BW25113 was deleted, generating the glucose-utilizing strain Bglc1. The glycerol dehydrogenase and dihydroxyacetone kinase-mediated glycerol dissimilation pathway was kept intact because it works mainly under anaerobic conditions[36]. Because strain Bgly1 and Bglc1 use separate carbon sources for growth and do not require essential ingredients from each other, their coculture is regarded as neutralistic.

We determined the growth profiles of Bgly1 and Bglc1 monocultures. The two strains showed specific utilization of glycerol and glucose, respectively. Bgly1 exhibited an exponential growth phase till 20 h in glycerol minimal media while Bglc1 quickly reached the stationary phase at 10 h in glucose minimal media (Supplementary Fig. 1). The two strains were transformed respectively with plasmids pSA-eGFP and pSA-mcherry and were cocultured at different IIRs (80%, 50%, and 20%, i.e., 4:1, 1:1, and 1:4; the same below). Cell growth was monitored by spectrometry and the population was determined by flow cytometry. At the population level, the apparent growth profile of the coculture is similar to the Bglc1 monoculture, and would reach the stationary phase in 12 h; the total cell density (measured by $OD_{600}$) was positively related to the IIRs of Bgly1 (Fig. 2a). At the strain level, Bglc1 dominated in the coculture, and its final percentage at 48 h ranged from 93.7 to 82.2% at the three IIRs (Fig. 2a). Both cell growth and population composition are affected by the IIRs, indicating the necessity to further improve the coculture stability.

To this end, a commensalistic coculture was constructed based on the above neutralistic coculture. Amino acids are frequently used as cross-feeding metabolites, and the biosynthetically costly ones such as methionine, lysine, and aromatics were shown to promote stronger correlation[27]. However, the single-metabolite correlation is relatively loose and insufficient to maintain stability under changing culture conditions. Further, many amino acid pairs are unable to sustain the normal growth of the cocultures due to their low leak into the extracellular environment[27]. Instead, we created a multi-metabolite correlation by engineering the glutamate node. Glutamate plays a central role in amino acid metabolism, and is the common amino doner for amino acid biosynthesis. E. coli synthesizes glutamate from ammonia via two

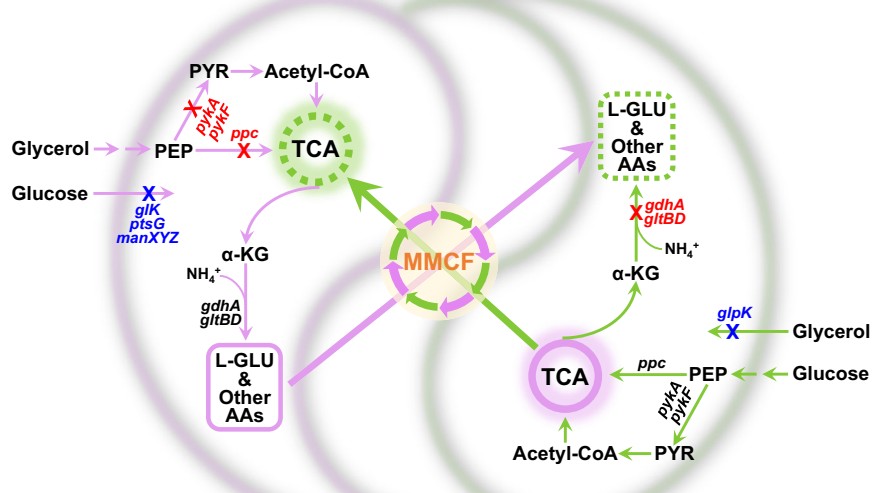

**Fig. 1 Design principles of the stable coculture system.** The two strains are engineered to utilize glycerol and glucose, respectively, and cross-feed each other with amino acids and the TCA cycle intermediates. Blue crosses denote gene knockouts to block glucose/glycerol utilization; red crosses denote gene knockouts to block glutamate biosynthesis and the TCA cycle. MMCF multi-metabolite cross-feeding. Genes: *glK* encodes glucokinase, *ptsG* encodes PTS glucose transport protein, *manXYZ* encodes mannose permease, *pykA/F* encodes pyruvate kinase, *ppc* encodes phosphoenolpyruvate carboxylase, *glpK* encodes glycerol kinase, *gdhA* encodes glutamate dehydrogenase, *gltBD* encodes glutamate synthase. Metabolites: PEP phosphoenolpyruvate, PYR pyruvate, α-KG α-ketoglutarate, L-GLU L-glutamate, AAs amino acids.

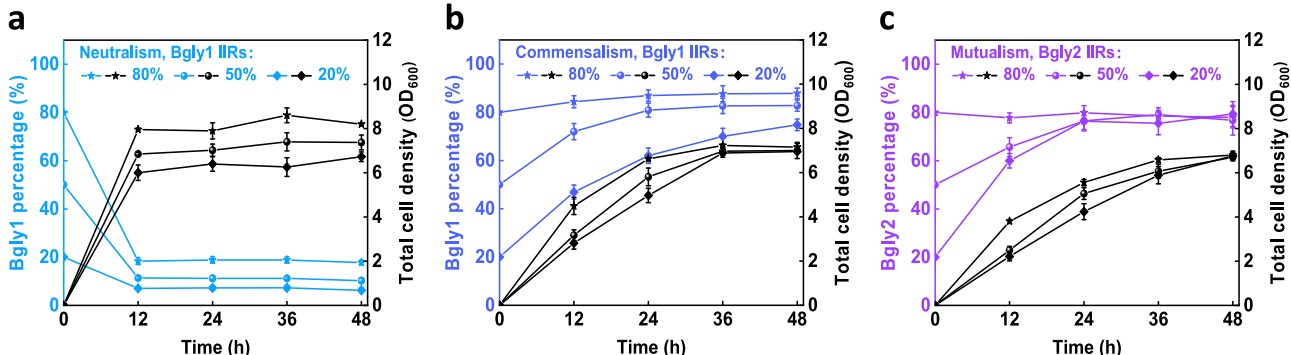

**Fig. 2 Curves of cell growth and population composition in the three coculture systems. a** The Neutralistic coculture; **b** the Commensalistic coculture; **c** the Mutualistic coculture. IIRs, the initial inoculation ratios. Data shown are mean ± SD ($n = 3$ independent experiments). Source data are provided as a Source Data file.

pathways catalyzed by glutamate dehydrogenase (encoded by *gdhA*) and glutamate synthase (encoded by *gltBD*), respectively. As these genes were deleted in strain Bglc2, it was unable to grow in glucose minimal media. Supplement of glutamate recovered growth in a dose-dependent manner, and supplement of other amino acids such as phenylalanine and tyrosine partially recovered cell growth via the endogenous transamination (Supplementary Fig. 2a and 2b). With this design, the amino acid metabolic networks of strains Bgly1 and Bglc2 would be connected, forming multi-point interactions. Because Bglc2 relies on amino acids from Bgly1 for growth while Bgly1 can grow independently, the Bgly1/Bglc2 coculture is regarded as commensalistic.

Encouragingly, we noticed that the total cell density of the Bgly1/Bglc2 coculture converged to around 7 at the three different IIRs (Fig. 2b). Bgly1 becomes the dominant strain, indicating that the growth of Bglc2 is limited by the supply of amino acids from Bgly1. Although the final ratios of Bgly1 tended to converge, they still ranged from 74.8 to 87.8% at 48 h (Fig. 2b).

The results indicate that the stability of the commensalistic coculture is not significantly improved compared with the neutralistic coculture, likely because strain Bgly1 is lack of restriction.

To strengthen the correlation between the members, we further constructed a mutualism coculture. Besides amino acid anabolism, energy metabolism is another essential function to support cell growth. The TCA cycle, coupling with the oxidative respiratory chain, is the major energy source under aerobic conditions. In addition, it contains 3 out of the 12 precursor metabolites that connect cell catabolism and anabolism (oxaloacetate, α-ketoglutarate, succinyl-CoA), and the involved carboxylic acids can transfer across cell membranes. Thus, the TCA cycle was selected as a second cross-feeding branch. As *pykA* and *pykF* were already deleted in Bgly1, gene *ppc* was further deleted to block the carbon flux to the TCA cycle, yielding strain Bgly2. As expected, Bgly2 was unable to grow in glycerol minimal media, and the growth can only be fully covered by co-addition of multiple carboxylic acids (Supplementary Fig. 3).

In the Bgly2/Bglc2 coculture, Bgly2 restored growth, confirming that Bglc2 can provide the TCA cycle metabolites. With such a multi-metabolite cross-feeding design, not only the total cell growth but also the population composition converged over time, and the percentages of strain Bgly2 were stabilized at around 76% in 48 h (Fig. 2c). Taken together, we can see that establishing a close multi-metabolite mutualistic relationship between the strains is essential to constructing a stable coculture system.

To determine the metabolites involved in cross-feeding, we performed the extracellular metabolomic analysis of the neutralistic and the mutualism cocultures. A total of 5 TCA cycle intermediates and 9 amino acids were detected in the supernatants of both cocultures, among which the levels of α-ketoglutarate and glutamate were significantly higher in the mutualistic coculture than in the neutralistic coculture (Supplementary Fig. 4). Considering the essential roles of α-ketoglutarate and glutamate in the TCA cycle and amino acid metabolism, we infer that they may serve as the key metabolites for crossing-feeding while the other relevant metabolites (carboxylic acids in the TCA cycle and amino acids) detected and even non-detected can also contribute to the cross-feeding.

**The stable culture system enables efficient salidroside biosynthesis**. To investigate the applicability in actual production, we compared the three coculture systems (neutralistic, commensalistic, and mutualistic) for heterologous chemical biosynthesis. Salidroside, an active plant natural product, was previously used as a demonstration of an *E. coli* single cross-feeding coculture system. For better comparison, we chose it as the first target compound. The aglycone tyrosol is synthesized via the shikimate pathway with tyrosine as a precursor, and is further glycosylated to salidroside with UDP-glucose as a glucosyl donor (Supplementary Fig. 5). Accordingly, the pathway was divided and allocated to the members of the cocultures, where the glycerol-utilizing strains are responsible for the upstream tyrosine biosynthesis while the glucose-utilizing strains for the downstream conversion of tyrosine to salidroside (Fig. 3a).

The relative stability of the cocultures was maintained after loading with the biosynthetic pathway, and the mutualistic coculture Bgly2-Tyr/Bglc2-Sal was the most stable with different IIRs. In the neutralistic coculture, the upstream strain Bgly1-Tyr took a minor percentage (10.5–21.4%), which limits the precursor supply and so does the final salidroside titers (337–683 mg/L) (Fig. 3b, e). In the commensalistic coculture, Bgly1-Tyr dominated throughout the cultivation process and its final ratios ranged from 89.5 to 73.2% (Fig. 3c); the salidroside titers ranged from 759 to 1276 mg/L at 48 h. (Fig. 3e). In the mutualistic coculture, the total cell growth, population composition, and salidroside titers were all strikingly stable with different IIRs. Although the IIRs varied from 4:1 to 1:4, the average salidroside titers at 48 h were stabilized at 1552 ± 27 mg/L, and the final percentages of Bgly2-Tyr converged at around 75% at 48 h (Fig. 3d and e). Interestingly, the amount of glycerol consumed (4.11 g/L at 48 h) was much less than that of glucose (9.69 g/L at 48 h) (Supplementary Fig. 6a), indicating that a significant amount of carbon flux derived from glucose is transferred from Bglu2-Sil to Bgly2-Tyr.

To explore the long-term stability of the mutualistic coculture, we conducted continuous passage cultivation in shake flasks. Every 24 h, the cell culture was taken as the seed and inoculated into the next fresh medium. The result showed that the coculture can maintain good production and population stability for up to 10 days (Supplementary Fig. 7). Interestingly, the titer even showed a gradual increase with the passage subcultures, indicating that the coordination between the counterparts may be further improved during the cultivation.

One feature of artificial microbial consortia is the division of labor. To investigate whether the mantainance of two populations causes the loss of production yield, we compared the mutralistic coculture with the monoculture for salidroside production. The result showed that the yield by the coculture (0.11 g/g) is equal to that by the monoculture, suggesting that the coculture does not necessarily sacrifice the carbon yield (Supplementary Fig. 6b). The slightly lower titer by the coculture (1523 mg/L versus 1603 mg/L) is attributed to the lower cell density (7.35 versus 8.56 at 48 h) (Supplementary Fig. 6c and d).

To evaluate the scalability of the mutualistic coculture, salidroside production was performed in 3 l bioreactors at a IIR of 50%. The percentage of strain Bgly2-Tyr declined rapidly to 30.3% during the first 12 h, then recovered to 73.2% at 24 h, maintained stable till 48 h, and finally slightly declined to 68.9% at 84 h (Fig. 3f). The general stability was maintained during the cultivation period. The initial perturbation should be attributed to that the nutritions in the seed culture (yeast extract and peptone) promote the growth of strain Bglc2. As for salidroside production, the titer increased continuously to 12.52 g/L at 84 h, with the yield of 0.12 g/g total carbon sources and the productivity of 0.15 g/L/h. The titer and productivity are doubled and tripled compared to those resulting in the previous study[10], respectively. The results showed that the mutualistic coculture could tolerate disturbance and maintain stability at bioreactor level, paving the way for its industrial application.

**Introducing a metabolite-responsive biosensor increases system tunability**. Besides stability, a versatile coculture system should also possess tunability to autonomously balance metabolic fluxes between the partial pathways distributed in different strains. To show the advantage of tunability, we chose coniferol biosynthesis as the second example. Coniferol is also synthesized via the shikimate pathway[37]. Briefly, tyrosine is converted to caffeate by tyrosine ammonia lyase (TAL) and 4-hydroxyphenylacetate 3-hydroxylase (HpaBC), and caffeate is then converted to coniferol viasequential reactions catalyzed by *p*-coumarate-CoA ligase (4CL1), caffeoyl-CoA O-methyltransferase (CCoAOMT), cinnamoyl-CoA reductase (CCR), and alcohol dehydrogenase (ADH) (Supplementary Fig. 5). The biosynthetic pathway was divided and introduced into strains Bgly2 and Bglc2, generating strains Bgly2-Caf and Bglc2-Con (Fig. 4a) In the Bgly2-Caf/Bglc2-Con coculture, the population percentages of strain Bglc2-Con ranged from 10 to 16.5% at 48 h (Fig. 4b). Coniferol was produced at low titers (62, 85, and 95 mg/L) with the accumulation of caffeate (175, 140, and 110 mg/L) at the three IIRs (Fig. 4c), indicating that the downstream strain Bglc2-Con is incapable of completing the conversion. Thus, it is necessary to modulate the strain population according to the actual demand of the pathway. As shown in Supplementary Fig. 2a and 2b, the population composition can be adjusted by supplementing ingredients such as glutamate or yeast extract. However, the doses need to be manually titrated and oftentimes is difficult to control precisely in large-scale production. Alternatively, the population composition can be modulated by fine-tuning the expression of *gdhA* in strain Bglc2-Con, but the static control using constitutive promoters may not match the dynamic change of cell metabolism during the cultivation process. Instead, we proposed to control *gdhA* expression with a caffeate-responsive biosensor, through which the accumulation of caffeate would therefore activate *gdhA* expression and promote the growth of strain Bglc2-Con. However, the implementation was hindered by the lack of known caffeate-responsive biosensors. A set of biosensors for aromatic compounds have been characterized[38–40]. According to their detection spectrum, the phenol-responsive DmpR biosensor and the salicylate-responsive NahR biosensor were inferred to be promising

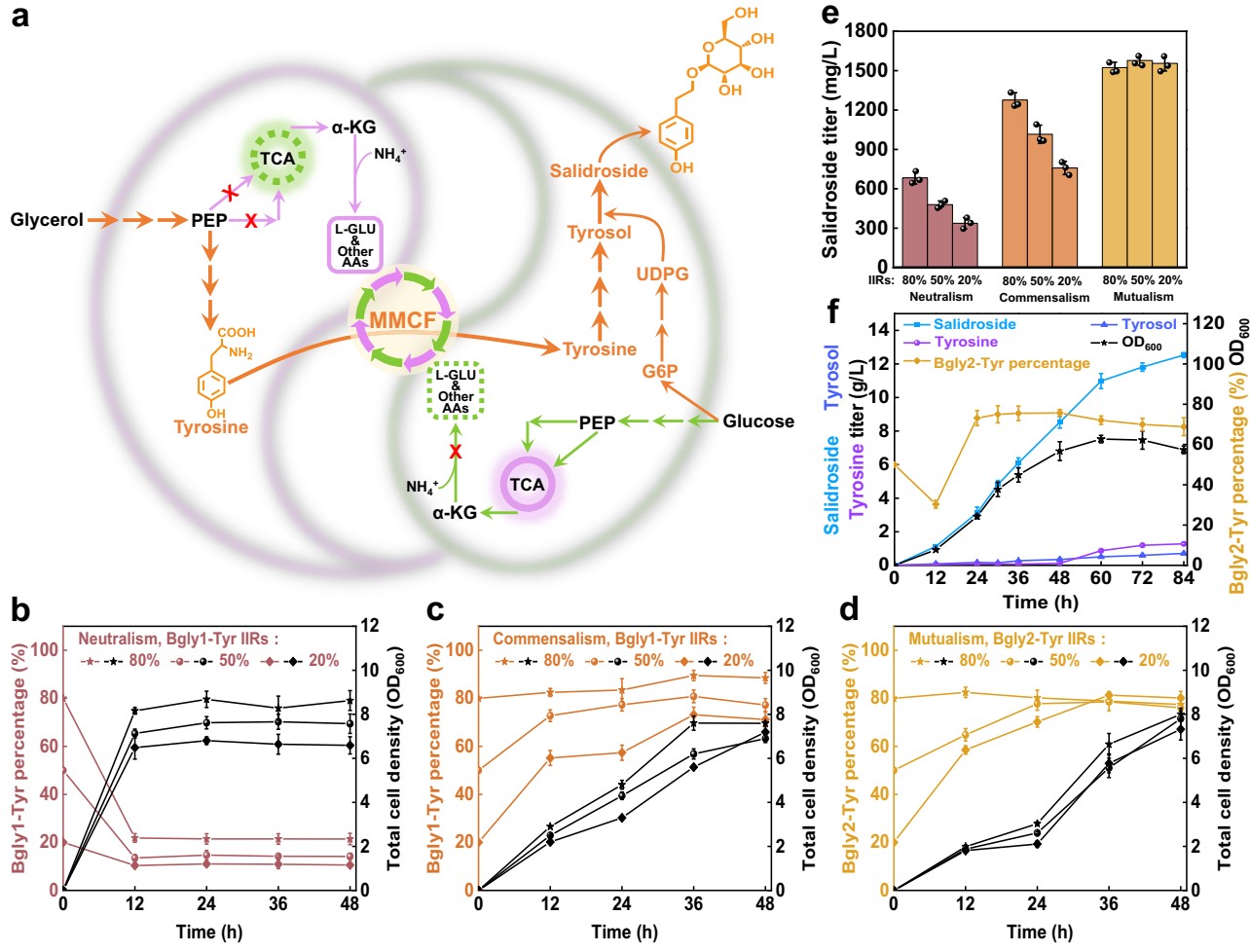

**Fig. 3 Efficient salidroside production and scale-up using coculture engineering. a** Schematic of the *E. coli-E. coli* cocultures to accommodate the salidroside biosynthetic pathway and convert a glycerol and glucose mixture to salidroside (The salidroside biosynthetic pathway is shown in orange. For the detailed pathway, see Supplementary Fig. 5). Curves of cell growth and population change in **b** the Neutralistic Bgly1-Tyr/Bglc1-Sal coculture, **c** the Commensalistic Bgly1-Tyr/Bglc2-Sal coculture, and **d** the Mutualistic Bgly2-Tyr/Bglc2-Sal coculture. **e** Comparison of salidroside titers in the three coculture systems. **f** Scale-up production of salidroside using the Bgly2-Tyr/Bglc2-Sal coculture. MMCF multi-metabolite cross-feeding, IIRs the initial inoculation ratios. Metabolites: PEP phosphoenolpyruvate, G6P glucose-6-phosphate, UDPG uridine diphosphate glucose, α-KG α-ketoglutarate, L-GLU L-glutamate, AAs amino acids. Data shown are mean ± SD (*n* = 3 independent experiments). Source data are provided as a Source Data file.

candidates. Their responsiveness was verified and assessed using a fluorescence reporter system. In brief, the transcriptional factors NahR and DmpR were constitutively expressed by the J23101 promoter, and the *egfp* gene was controlled by their cognate promoters. The DmpR biosensor showed expected responsiveness to caffeate with a 12-fold increase in fluorescence intensity at 2 mM caffeate than the control, while the NahR did not (Supplementary Fig. 8a and 8b). In comparison, 0.25 mM phenol led to a 50-fold increase (Supplementary Fig. 8c), indicating that caffeate has a lower affinity to DmpR than phenol. Besides, the DmpR biosensor showed only weak responsiveness (less than one-fold) to other pathway intermediates (L-dopa, tyrosine, and *p*-coumaric acid) (Supplementary Fig. 8d).

Considering the suitable sensitivity and specificity, the DmpR biosensor was used to control *gdhA* expression. GdhA was fused with a proteinase-degradation-tag SsrA to timely shutdown the response when the triggering signal disappears[41,42]. Strain Bglc2-Con was transformed with plasmid pSA-P_dmp-GS-PJ23101-dmpR, generating strain Bglc2-ConDmpR. In the Bgly2-Caf/Bglc2-ConDmpR coculture, the percentage of the downstream strain was increased from 16 to 52%, and the average total cell density (OD_600) also increased from 5.9 to 7.8 (Fig. 4d).

Consequently, coniferol titers were significantly improved (217, 258, and 246 mg/L) when only a small amount of caffeate accumulated (20, 5, and 4 mg/L) at the three IIRs (Fig. 4e). The results demonstrated that the population composition could be self-regulated while the stability is still maintained.

**Establishing a three-strain coculture improves the production of silybin/isosilybin.** To further explore the application potential, we used the stable and tunable coculture system for de novo production of the structurally more complicated compounds silybin and isosilybin, which have been used as hepatoprotectives[43,44]. They are biosynthesized via a divergent-convergent pathway (Supplementary Fig. 5). Two precursors (coniferol and taxifolin) are both derived from caffeate and are condensed to form silybin/isosilybin catalyzed by the ascorbate peroxidase APX1[45]. Starting from tyrosine, the pathway involves 10 enzymatic steps. A two-step enzymatic cascade has been constructed for efficient biotransformation of eugenol and taxifolin to silybin/isosilybin[46]. Recently, the production of silybin/isosilybin has been achieved by combining metabolic engineering approaches with enzymatic catalysis[47]. In brief, the two precursors were separately synthesized from glucose using two

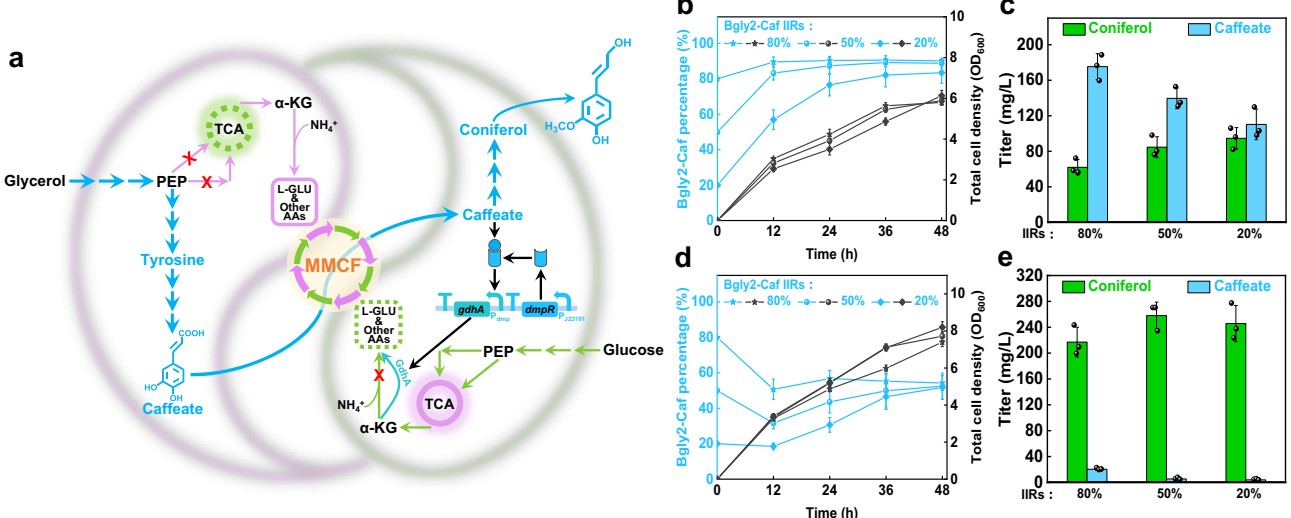

**Fig. 4 Production of coniferol using coculture engineering. a** Schematic of the *E. coli-E. coli* cocultures to accommodate the coniferol biosynthetic pathway and convert a glycerol and glucose mixture to coniferol (The coniferol biosynthetic pathway is shown in sapphire. For the detailed pathway, see Supplementary Fig. 5). **b** Curves of cell growth and population change in the static Bgly2-Caf/Bglc2-Con coculture. **c** Titers of coniferol and caffeate at 48 h in the Bgly2-Caf/Bglc2-Con coculture. **d** Curves of cell growth and population change in the dynamic Bgly2-Caf/Bglc2-ConDmpR coculture. **e** Titers of coniferol and caffeate at 48 h in the Bgly2-Caf/Bglc2-ConDmpR coculture. MMCF multi-metabolite cross-feeding, IIRs the initial inoculation ratios. Genes: *gdhA* encodes glutamate dehydrogenase, *gltBD* encodes glutamate synthase, Enzymes: GdhA glutamate dehydrogenase, DmpR the caffeate-responsive transcription factor. Metabolites: PEP phosphoenolpyruvate, α-KG α-ketoglutarate, L-GLU L-glutamate, AAs amino acids. Data shown are mean ± SD (*n* = 3 independent experiments). Source data are provided as a Source Data file.

metabolically engineered *S. cerevisiae* strains. After purification they were further converted to silybin/isosilybin by APX1. However, their one-pot de novo biosynthesis has not yet been achieved, suggesting the great challenges in modulating the full pathway.

The modularity of coculture engineering can simplify the construction and optimization process. As for silybin/isosilybin biosynthesis, the above caffeate-producing strain Bgly2-Caf can be used directly, and thus the focus was on the integration of the downstream modules. To guide the pathway assembly and optimization, the conversion capacities of the two branches were first evaluated. Two plasmids pZE-CCF4 and pZE-CA4C were constructed and used for the feeding experiments. When fed with 500 mg/L caffeate, the corresponding strains produced 268 mg/L of coniferol and 8 mg/L of taxifolin in 48 h, respectively (Supplementary Fig. 9), indicating that the taxifolin branch is rate-limiting. In addition, the peroxidase APX1 requires $H_2O_2$ as the co-substrate, and the NADH oxidase Nox from *Lactococcus lactis* was co-expressed to enable endogenous supply of $H_2O_2$[48]. Accordingly, these modules were accommodated on plasmids pZE-CA, pCS-$P_{lac}$-CF4C-$P_{J23101}$-CA, pSA-$P_{dmp}$-gdhA-$P_{J23101}$-DN. The plasmids were co-transferred into strain Bglc2, generating strain Bglc2-Sil. The coculture of Bglc2-Caf/Bglc2-Sil produced 0.28 mg/L of silybin/isosilybin in 48 h with the accumulation of 101 mg/L caffeate and 9 mg/L coniferol (Fig. 5 b and c). The accumulation of caffeate did not further stimulate the growth of Bglc2-Sil, probably due to its heavy metabolic load (Fig. 5d).

To reduce the metabolic burden and competition, the downstream pathway was distributed into two Bglc2 cells. The two strains Bglc2-Con(b) and Bglc2-Sil(b)were cocultured with the upstream strain Bglc2-Caf, forming a three-strain coculture (Fig. 5a). Silybin/isosilybin titers increased by six fold to 2.02 mg/L, and caffeate accumulation was reduced to 71 mg/L (Fig. 5b, c). In addition, the percentage of Bglc2-Sil(b) kept increasing to 72% within 48 h (Fig. 5e), showing the effectiveness of the biosensor. On the other hand, the percentage of Bglc2-Caf

remained stable at around 18% after 12 h while that of Bglc2-Con(b) (without the biosensor) began to decrease after 24 h and dropped to 9% at 48 h (Fig. 5e). Successful biosynthesis of silybin/isosilybin demonstrates the potential of the dynamic mutralistic coculture for biosynthesis of complex chemicals.

## Discussion

Microbial consortia are widespread in nature and play important roles in maintaining ecological balance and human health[7,9,49,50]. The interactions of the strains in natural microbial consortia are highly dynamic and complex, and are inherently challenging to investigate. In recent years, artificial microbial consortia are harnessed for biomanufacturing and studying cell-cell interactions. Coculture engineering shows a promising application in the biosynthesis of complex chemicals.

Stability and tunability are two desirable properties of a coculture system. In this study, the two properties were realized simultaneously. The resulting cocultures can, on one hand, be used as promising platforms for biosynthesis; on the other hand, as simplified models, they would help unravel the interaction and regulation mechanisms of natural microbial communities. To achieve stability, we established a MMCF strategy to strengthen the correlation between cocultured strains. The corresponding criteria were also proposed as the essentiality, diversity, and transferability of the involved metabolites. Besides the TCA cycle and the amino acid anabolism, other metabolic branches such as the shikimate pathway and the asparate superpathway, also meet the criteria. By combinatorially using these branches, a series of stable cocultures can be generated, demonstrating the versatility of this strategy. To achieve tunability, we integrated the biosensors into the coculture system for autonomously regulating the population composition. In our cases, only the growth of the downstream strain was up-regulated to achieve pathway balance. When necessary, this strategy can be readily extended to achieve bidirectional oscillating population control by integrating with CRISPRi or RNAi. To validate the metabolites that may

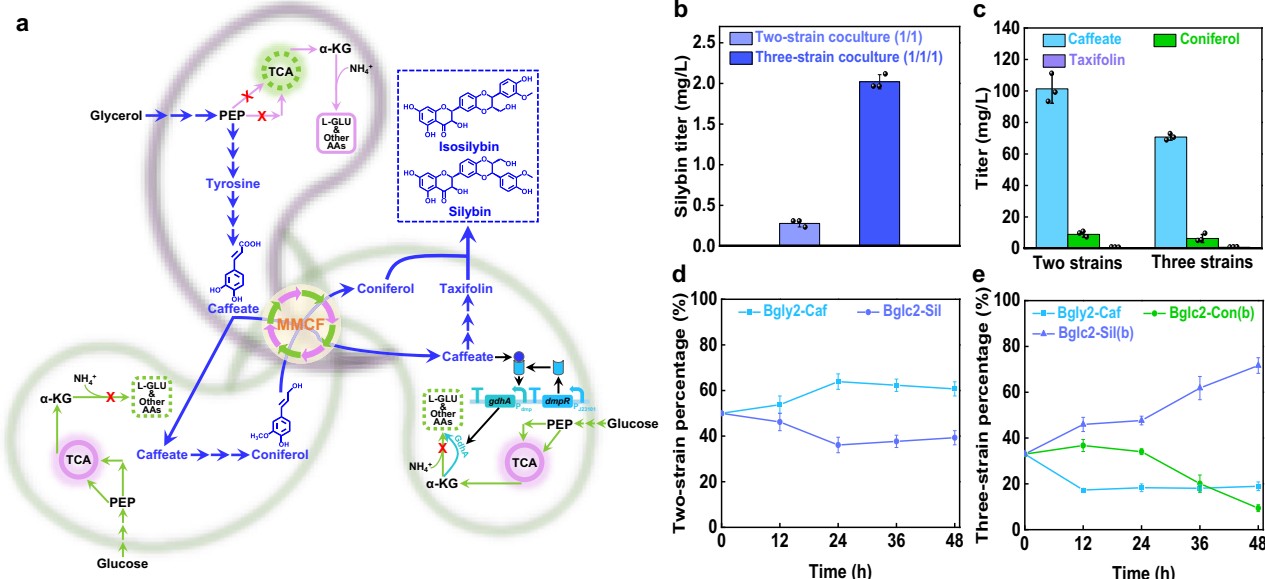

**Fig. 5 Production of silybin/isosilybin using coculture engineering. a** Schematic of the three-strain coculture to accommodate the silybin/isosilybin biosynthetic pathway and convert a glycerol and glucose mixture to silybin/isosilybin (The silybin/isosilybin biosynthetic pathway is shown in blue. For the detailed pathway, see Supplementary Fig. 5). **b** Silybin/isosilybin titers. **c** Amounts of the intermediates accumulated. **d** Population change of the two-strain coculture. **e** Population change of the three-strain coculture. MMCF multi-metabolite cross-feeding. Genes: *gdhA* encodes glutamate dehydrogenase, *gltBD* encodes glutamate synthase. Enzymes: GdhA glutamate dehydrogenase, DmpR the caffeate-responsive transcription factor. Metabolites: PEP phosphoenolpyruvate, α-KG α-ketoglutarate, L-GLU L-glutamate, AAs amino acids. Data shown are mean ± SD (*n* = 3 independent experiments). Source data are provided as a Source Data file.

participate in the crossing-feeding, we performed a metabolomic analysis of the culture supernatants. In the future, the application of isotopic metabolic flux analysis would be greatly helpful to give a clearer map about growth dynamics and population interaction within the cocultures[51,52]. As for the production yield, we observed that the specialized coculture cells may not lead to the sacrifice of carbon yield. Thus, the coculture strategy may find promising applications in the biosynthesis of chemicals with long biosynthetic pathways or complex structures. However, for the biosynthesis of relatively simple chemicals, the monoculture strategy is still the priority.

In addition to the homogenous coculture, this strategy can also be expanded to the heterogeneous coculture. For instance, pro-karyotes such as *E. coli* have faster growth rates and are favored to efficiently produce skeleton molecules[53,54] while eucaryotes such as *S. cerevisiae* have abundant inner membrane structures and advanced post-translational modification mechanisms, suitable for expressing tailoring enzymes such as cytochrome P450s[55]. Developing heterogeneous cocultures can take advantage of different species for the efficient synthesis of complex chemicals.

## Methods

**Strains and plasmids**. Strains and plasmids used in this study are listed in Supplementary Table 1 and Supplementary Table 2, respectively. Primers designed in this study for gene amplification or plasmid construction are listed in Supplementary Data 1. *E. coli* strain DH5α was used as the host strain for plasmid construction. *E. coli* strain BW25113 and its derivatives were used for the biosynthesis of the target products[56]. Plasmids pZE12-luc (high copy), pCS27 (medium copy), and pSA74 (low copy) were employed for pathway assembly[56]. The plasmids were constructed using the standard enzyme digestion and ligation method. Gene knockout was carried out using RED recombination following the standard protocols[57].

**Culture media and conditions**. Luria-Bertani (LB) medium containing 10 g/L tryptone, 10 g/L NaCl and 5 g/L yeast extract was used for strain maintenance and seed propagation. All batch fermentations, including monocultures and cocultures, were carried out in M9 media containing 6.78 g/L Na$_2$HPO$_4$, 3 g/L KH$_2$PO$_4$, 1 g/L NH$_4$Cl, 0.5 g/L NaCl, 1 mM MgSO$_4$, 0.1 mM CaCl$_2$, 15 g/L glucose and/or 15 g/L

glycerol. Trace elements were also supplemented to all batch fermentations, giving final concentrations of 0.03 mg/L H$_3$BO$_3$, 0.38 mg/L CuCl$_2$, 0.4 mg/L Na$_2$EDTA, 0.5 mg/L CoCl$_2$, 0.94 mg/L ZnCl$_2$, 1.6 mg/L FeCl$_2$, 3.6 mg/L MnCl$_2$, and 100 mg/L thiamine. Ampicillin, kanamycin, and chloramphenicol were added to the media when needed at final concentrations of 100, 50, and 34 mg/mL, respectively.

Shake flask experiments were performed at 37 °C and 220 rpm. Overnight seed cultures were diluted 1:50 into 50 mL fresh M9 media, and induced with 0.5 mM isopropy-soprthiogalactoside (IPTG) at 3 h after inoculation. Samples were taken every 12 h. Cell growth was monitored by measuring the optical density at 600 nm (OD$_{600}$) and product accumulation was analyzed with high-performance liquid chromatography (HPLC). To validate the metabolites involved in cross-feeding, the supernatants of the neutral (Bgly1/Bglc1) and the mutualistic (Bgly2/Bglc2) cocultures were taken at 12 h and 36 h, centrifuged, and subjected to metabolic analysis by Shanghai Applied Protein Technology, Ltd.

For scale-up production of salidroside, strains Bgly2-Tyr and Bglc2-Sal were inoculated with a ratio of 1:1 into 3 L bioreactors containing 1 L M9 media. The dissolved oxygen (DO), pH, and temperature were automatically maintained at 15% (v/v), 7.0, and 37 °C. The feeding solution contained 150 g/L glucose and 150 g/L glycerol. Glucose concentration was maintained below 5 g/L. Samples were taken at regular time intervals for analysis of cell growth and product accumulation. The data were analyzed by the software OriginPro 9.0.

**Determination of population composition in the cocultures**. Green or red fluorescent protein genes were transferred into the strains and the number of cells with different fluorescence in the samples was counted by flow cytometry (Beckman CytoFlexS, USA) equipped with three lasers (405 nm, 488 nm, 635 nm) and 6 fluorescence channels (FITC, PE, PerCP, APC, PE-Cy5, PE-Cy7). Forward Scatter (FSC, reflecting the volume size and viability of the tested cells, 488/8) and Side Scatter (SSC, reflecting the internal cell granularity, 488/8) were utilized synergistically to screen eligible strains. The total number of assays was set to 100,000, the sample flow rate was medium (30 μL/min), and the cell granularity threshold was >800. For strains with green fluorescence, SSC and Fluorescein Isothiocyanate (FITC, 525/40) filters were utilized synergistically. To screen for strains with red fluorescence, SSC and Phycoerythrin Cyanin 5 (PE-Cy5, 690/50) filters were utilized synergistically. Samples were diluted 1000-fold and placed in 2 ml collection tubes for loading assays, and population ratios were calculated based on the number of red, green or empty cells detected using the software CytExpert 2.0. Two examples of the flow cytometry results were shown in Supplementary Fig. 10.

**Construction of caffeate biosensors**. To test the response of the DmpR and NahR biosensors to caffeate and other compounds, the transcription factor genes

were controlled by the constitutive promoter $P_{J23101}$ on plasmids pSA-$P_{J23101}$-dmpR and pSA-$P_{J23101}$-nahR. The green fluorescent reporter gene *egfp* was controlled by the cognate promoters on plasmids pZE-$P_{dmp}$-eGFP and pZE-$P_{nah}$-eGFP. *E. coli* BW25113 was transformed with the plasmids pairs, respectively, and the profiles of fluorescence intensities at different inducer concentrations were monitored with microplate reader. To regulate strain growth in response to caffeate, plasmid pSA-$P_{J23101}$-dmpR-$P_{dmp}$-gdhA-ssrA was constructed, in which *gdhA* is fusion-expressed with gene *ssrA* (encoding a proteinase-degradation tag AANDENYALAA).

**HPLC analysis**. Tyrosol, salidroside, caffeate, coniferol, silybin/isosilybin and other metabolites were analyzed and quantified by reverse phase HPLC (HITACHI) equipped with a Diamonsil C18 column (Diamonsil 5 μm, 250 × 4.6 mm). Both the standards and samples were centrifuged at 12,000 rpm for 2 min, and the supernatant was filtered through a 0.22 μm organic filter membrane. Solvent A was water with 0.1% formic acid and solvent B was methanol. The column temperature was set to 40 °C. The following gradient was used at a flow rate of 1 mL/min: 10 to 70% solvent B for 15 min, 70 to 100% solvent B for 4 min, 100 to 10% solvent B for 1 min, and 10% solvent B for an additional 5 min. Quantification was based on the peak areas at specific wavelengths (276 nm for tyrosine, tyrosol, and salidroside, 323 nm for caffeate, 260 nm for coniferol, 280 nm for taxifolin and silybin/isosilybin). The production of silybin/isosilybin was further confirmed by mass spectrum analysis (Supplementary Fig. 11). The standard curve for each compound was provided in Supplementary Fig. 12.

**Reporting summary**. Further information on research design is available in the Nature Research Reporting Summary linked to this article.

## Data availability
Data supporting the findings of this work are available within the paper and its Supplementary Information files. A reporting summary for this Article is available as a Supplementary Information file. Source data are provided with this paper.

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

## Acknowledgements

This work was supported by the National Key Research and Development Program of China (2021YFC2100800) and the National Natural Science Foundation of China (21978015, 21776008, and 21636001).

## Author contributions

XX.S. and Q.Y. designed the experiments. X.L. performed the experiments. Z.Z. and W.L. constructed plasmids. XL.S. and J.W. performed the fed-batch production experiment. X.L. and X.S. analyzed the data and wrote the manuscript. XX.S., Q.Y., XL.S., J.W., and Y.Y. participated in the discussion and revision of the manuscript.

## Competing interests

The authors declare no competing interests.
