## [Peer Review File · Nature Communications]

Design of stable and self-regulated microbial consortia for chemical synthesisReviewers' Comments:

Reviewer #1:

Remarks to the Author:

This paper described microbial co-culture engineering for biomanufacturing. By using cross-feeding, authors could establish robust co-cultures or tri-cultures where the final strain proportion and the product titer are insensitive to the initial inoculation ratios. The *gdhA* biosensor could also be used to tune the culture dynamics. This smart approach improved system stability and productions. The overall research is interesting and innovative. However, the authors still need to improve the paper quality to meet the high requirement from Nature Communication Journal. A few suggestions are following:

1. The pathways in Fig 1, 3 and 4 needs more details. I suggest three pathways can be combined to a more detailed map. Based on their pathway maps, I cannot understand how cells can produce acetyl-CoA (precursor of lipid and leucine) without *pykA*, *pykF* and *ppc*. By blocking fluxes to the TCA cycle, the *E.coli* metabolism can be severely disrupted since NADH, citrate and other TCA cycle metabolites regulates glycolysis enzymes. It is confusing to me how their *Bgly1* strain can still grow in the mon-culture? More explanations are needed for cell metabolism. Any flux balance analysis can be helpful.

2. Coculture aims to achieve the division of labor. However, I am not convinced to use coculture to produce salidroside. Since Tyr is a native amino acid and its biosynthesis pathway has low metabolic burden. Use one strain to overproduce Tyr seems not necessary. The authors should discuss how the co-culture productivity (including titer and rate) can be improved comparing to the mono-culture productivity (i.e., all pathways are placed in one strain). Particularly, to strengthen the correlation between the members, the strains innate pathway has to be removed that will cause cell slow growth. The authors need more guidelines about the Pros and Cons for coculture engineering.

3. In their scalability study on the mutualism co-culture, their 3-liter bioreactor batch operation could not give much information about strain's industrial applicability. Instead, the authors should run a chemostat fermentation and thus the long term continuous culture can determine the strain stability and scalability (>60 generations) (Current Opinion in Biotechnology 66, 227-235).

4. The cross-feeding may be limited by cell carboxylic acids secretions, metabolite transporters and accumulated metabolite concentrations. The authors could perform more metabolomic analyses in the supernatant. During the cultivation, did free metabolite concentrations maintain constant or fluctuate? Moreover, this paper did not talk about the carbon yields. The authors should report the substrate consumption curves. I am curious if coculture will cause the loss of production yield since the system needs to maintain two populations.

5. Green or red fluorescent protein are used to analyze total cell populations. This method may not determine the live population ratio. Can they use CFU counting to further confirm the population dynamics?

6. The authors may consider to add population kinetic models to explain and simulate the subpopulation interactions (e.g., Algal Research 58, 102372). The model can be helpful to explain how inoculation ratio and relative growth rates affect cell population dynamics.

7. The paper contains some typos and minor grammatical errors (e.g., Fig 4 missed a sub-fig caption g; line 409 miss a temperature symbol). The authors need to proofread the paper again.

Reviewer #2:

Remarks to the Author:

In this manuscript, authors engineered new microbial consortia based on multiple-metabolite cross-

feeding, and a salidroside-producing co-culture system was developed from it, with a yield of 12.52 g/L. Then, a caffeate-responsive biosensor was used in a coniferol-producing microbial consortia, which significantly increased the yield. Finally, a three-strain system was constructed for production of silybin and isosilybin. This study provides a new strategy for modular co-culture engineering. The MS is suggested for publication in Nature Communications after the below issues are addressed.

1. In the introduction, the authors described "However, the co-culture systems designed to date require adjustment of the initial inoculation ratios (IIRs) to optimize, severely limiting use for large-scale production". The relevant references should be cited to support this.
2. Still in the introduction, the authors described "Previous symbiotic relationships have been established primarily based on single-metabolite cross-feeding, with loose correlations between the strains that are prone to disruption". It needs more concrete data from the previous reports to support this point.
3. Again, in the introduction, the authors described "Self-regulation is another desired feature of a robust microbial co-culture, where the strain ratio is autonomously coordinated to maximize metabolic flux of the product with minimal build-up of the intermediates." Here, suggest to use a more conservative word to replace "maximize", because it is difficult to say a population ration of the co-culture strains established by biosensor (including the caffeate-responsive biosensor in this MS) is the best ration for compound production.
4. Please add more detailed description to define three co-culture systems (neutralism, commensalism and mutualism).
5. The abbreviation should be defined at the first place in the main text, for examples ptsG, manXYZ and glk. In this regard, it is strongly recommended to include more background of selected enzymes/genes in the metabolic engineering studies.
6. To accurately quantify the compounds, the standard curve of each compound should be plotted and must make sure the concentration of each sample for analysis is within the linear region.
7. In line 44, should reference 10 be reference 11?

Reviewer #3:

Remarks to the Author:

In this manuscript, Li et al. reported a few new strategies to design microbial consortia. Their main objective was to develop consortia that are less sensitive to the initial inoculation ratio (being more robust). To this end, they constructed two strains that could use glucose and glycerol, respectively. They found that the glucose-utilizing strain (Bg1c1) grew much faster than the glycerol strain (Bg1y1) when being co-cultured in a medium containing both glucose and glycerol. This led to Bg1c1 dominating the final population. To balance the ratio of the two strains, they engineered Bg1c1 so that it cannot synthesize glutamate from alpha-ketoglutarate using ammonium or glutamine as the amino donor (new strain: Bg1c2); they modified Bg1y1 so that it cannot synthesize any TCA intermediates (new strain: Bg1y2). They confirmed that Bg1c2 and Bg1y2 could 1) exist in a balanced ratio when co-cultured in the medium and 2) be used to synthesize salidroside efficiently. They further introduced biosensors into the co-culture for improving synthesis of new target molecules. They also expanded the system to demonstrate that a three-member cross-feeding co-culture could also be constructed using a similar concept.

I think the reported method is interesting and potentially better than existing ones. I wish to request the authors to address the following concerns to improve their manuscript:

1. Is it necessary to use such complicated design? The authors needed to introduce many cross-feeding designs to balance the two subpopulations. A reason is that Bglc1 grew much faster than Bgly1. When Zhang et al. reported their glucose/xylose co-culture in 2015 (PNAS), it seems that they did not encounter a similar problem, possibly because their glucose-utilizing strain and xylose-utilizing strain had similar growth rate. It would be helpful if the authors could test salidroside production using the glucose/xylose strategy, which can be considered as a commonly used strategy in this field. The comparison would help assess the value of the new strategy being reported here. Bgly1 can utilize xylose but not glucose, so it can be used as the xylose strain. It should be straightforward to disable xylose utilization to get a new glucose strain for the glucose/xylose co-culture.

2. What are the metabolite exchanges Bgly2 had with Bglc2 when they are co-cultured in the glucose/glycerol medium? The authors only mentioned that co-culturing with Bglc2 restored the growth of Bgly2 but did not specify the details of the interactions. A further investigation at the molecular level would substantially improve the depth of this study. There are many interesting questions that can be asked. For example, what are the key metabolites being transported from Bglc2 to Bgly2? Is it required to transport many TCA intermediates or only one critical metabolite? The authors may consider to answer the question through 1) ¹³C- Metabolic Flux Analysis, 2) Transcriptome analysis, and/or 3) Perturbing the interaction through gene deletion/inactivation.

Reviewer #4:

Remarks to the Author:

In order to create robust biomanufacturing-focused co-cultures, the authors examine the effects of multi-metabolite cross-feeding on strain proportion and product titer. They begin by noting that the maintenance of co-cultures via single-metabolite exchange is prone to variability and sensitive to the initial inoculation ratio (IIR). They propose to address this by increasing the codependence of each strain using multi-metabolite cross-feeding. First, the authors analyzed how the serial implementation of this strategy (moving from neutralism to commensalism and then to mutualism) affected the final strain ratio given different IIRs. Then, they compared the same systems in their ability to efficiently synthesize salidroside after giving one half of the metabolic pathway to each constituent member. They also examined the effect of tuning these ratios by tying the growth of the less-dominant strain to the intermediate of an engineered coniferol pathway. Finally, they investigated the potential of this strategy to synthesize more complex molecules like silybin by comparing the biosynthetic efficiency of two- and three-strain implementations of this pathway.

The authors find that their chosen mutualistic relationship successfully holds their co-cultures to tunable strain ratios which can then be used to dramatically increase biosynthetic efficiency. They first find that a wide range of IIRs do not disrupt the system's final strain ratio. They also find that maintaining and tuning this ratio via intermediate sensing improves their ability to synthesize salidroside and coniferol, again at a wide range of IIRs. Finally, they find that they are able to improve this efficiency even further in more complex pathways by splitting up the biosynthesis of silybin into three-strains while tuning the growth of one of the less-dominant strains. Based on its clear presentation and strong improvement over cited alternatives, I believe this paper could represent a strong contribution to this journal with a few important revisions.

Chiefly, the authors argue throughout the paper that their knockouts of the ppc, gdhA, and gltBD genes represent a definite example of the exchange of multiple metabolites between two strains. However, they demonstrate this by serially adding missing nutrients in monoculture and examining the recovered growth. I would expect similar growth improvements (between single and multi-metabolite growth) in non-deficient cultures, as one would expect in the comparison of growth in rich vs. minimal media. Without a direct experimental comparison to non-deficient cultures (and ideally individual knockout cultures as well), it is difficult to say that these strains are actually exchanging all

of these metabolites. I would suggest either providing new evidence, as outlined above, or revising the claims made in this paper to make it clear that this multi-metabolite is a hypothetical explanation for the results seen in this paper.

I would also like to see the following addressed:

-In general, the conceptual metabolic pathway schematics are difficult to read. Figure 5a has so many overlapping lines in the center that it is almost impossible to interpret.

-Some more depth on the statistical analysis of this data would be appreciated.

-Besides their different biosynthetic pathways, what is the relationship between the two glucose-dependent strains of section 2.4? Are they missing the same nutrients? If they contain the same knockouts (and therefore require the same nutrients from the glycerol-dependent strain), this could possibly further undermine the hypothesis that multi-metabolite exchange is responsible for the improved efficiency reported here. This is especially concerning since the authors did not vary IIR in this final section.

We greatly appreciate the review comments. We have carefully revised the manuscript and highlighted all the changes in yellow in the revised manuscript.

REVIEWER COMMENTS

Reviewer #1 (Remarks to the Author):

This paper described microbial coculture engineering for biomanufacturing. By using cross-feeding, authors could establish robust cocultures or tri-cultures where the final strain proportion and the product titer are insensitive to the initial inoculation ratios. The *gdhA* biosensor could also be used to tune the culture dynamics. This smart approach improved system stability and productions. The overall research is interesting and innovative. However, the authors still need to improve the paper quality to meet the high requirement from Nature Communication Journal. A few suggestions are following:

1. The pathways in Fig 1, 3 and 4 needs more details. I suggest three pathways can be combined to a more detailed map. Based on their pathway maps, I cannot understand how cells can produce acetyl-CoA (precursor of lipid and leucine) without *pykA*, *pykF* and *ppc*. By blocking fluxes to the TCA cycle, the *E. coli* metabolism can be severely disrupted since NADH, citrate and other TCA cycle metabolites regulates glycolysis enzymes. It is confusing to me how their Bgly1 strain can still grow in the mono-culture? More explanations are needed for cell metabolism. Any flux balance analysis can be helpful.

Response: Thanks for the reviewer's suggestion. We revised Figures 1, 3, 4 and 5. The biosynthetic pathways of salidroside, coniferol, taxifolin and silybin were combined into one detailed map (Supplementary Fig. 5) while the simplified pathways were shown in Figures 3, 4 and 5. Figure 1 was revised to give a clearer and more detailed design schematic of the mutualistic coculture.

In strain Bgly1, genes *pykA* and *pykF* were knocked out, but gene *ppc* was still intact. Thus, it can still grow in the mono-culture. Strain Bgly2 with the *pykA/pykF/ppc* triple knockouts was unable to grow in the mono-culture. In the coculture, the growth of strain Bgly2 was recovered due to the acquirement of metabolites involved in the TCA cycle from the counterpart.

2. Coculture aims to achieve the division of labor. However, I am not convinced to use coculture to produce salidroside. Since Tyr is a native amino acid and its biosynthesis pathway has low metabolic burden. Use one strain to overproduce Tyr seems not necessary. The authors should discuss how the coculture productivity (including titer and rate) can be improved comparing to the mono-culture productivity (i.e., all pathways are placed in one strain). Particularly, to strengthen the correlation between the members, the strains innate pathway

has to be removed that will cause cell slow growth. The authors need more guidelines about the Pros and Cons for coculture engineering.

Response: Thanks for the reviewer's comments. Salidroside production was used as an initial example to better compare with the previous-designed coculture systems. Our system showed significant advantages in the stability and production.

We added the experiment of producing salidroside by a single strain. The result showed that the yield by the coculture (0.11 g/g) is equal to that by the monoculture, suggesting that the coculture does not necessarily sacrifice the carbon yield (Supplementary Fig 6b). The slightly lower titer by the coculture (1523 mg/L versus 1603 mg/L) is attributed to the lower cell density (7.35 versus 8.56 at 48 h) (Supplementary Fig. 6c and 6d). We added this data in the Result Section (Line 213-220) and gave a brief discussion in the Discussion Section (Line 348-356).

3. In their scalability study on the mutualism coculture, their 3-liter bioreactor batch operation could not give much information about strain's industrial applicability. Instead, the authors should run a chemostat fermentation and thus the long term continuous culture can determine the strain stability and scalability (>60 generations) (Current Opinion in Biotechnology 66, 227-235).

Response: Thanks for the reviewer's suggestion. Instead of running chemostat fermentation, we conducted continuous passage cultivation in shake flasks (Lines 206-212). Every 24 hours, 1 mL of the cell culture was taken as the seed and inoculated into 50 mL of the next fresh medium. The result showed that the coculture can maintain good production and population stability for up to 10 days (Supplementary Fig. 7). Interestingly, the titer even showed a gradual increase with the passage subcultures, indicating that the coordination between the counterparts may be further improved during the cultivation.

4. The cross-feeding may be limited by cell carboxylic acids secretions, metabolite transporters and accumulated metabolite concentrations. The authors could perform more metabolomic analyses in the supernatant. During the cultivation, did free metabolite concentrations maintain constant or fluctuate?

Moreover, this paper did not talk about the carbon yields. The authors should report the substrate consumption curves. I am curious if coculture will cause the loss of production yield since the system needs to maintain two populations.

Response: Thanks for the reviewer's suggestions. We performed the metabolomic analysis of the supernatants of the neutral and mutualistic cocultures (Lines 169-178, Supplementary Fig. 4). A total of 5 TCA cycle intermediates and 9 amino acids were detected in the supernatants of both cocultures, among which the levels of α -ketoglutarate and glutamate were significantly higher in the mutualistic coculture than in the neutralistic coculture. Considering the essential roles of α -ketoglutarate and glutamate in the TCA cycle and amino acid metabolism, we infer

that they may serve as the key metabolites for crossing feeding while the other relevant metabolites (carboxylic acids in the TCA cycle and amino acids) detected and even non-detected can also contribute to the cross-feeding.

For solidoside production, we provided the carbon yields and the substrate consumption curves (Supplementary Fig. 6a), and also compared the coculture with the monoculture. The result showed that the yield by the coculture (0.11 g/g) is equal to that by the monoculture, suggesting that the coculture does not necessarily sacrifice the carbon yield (Supplementary Fig 6b). The slightly lower titer by the coculture (1523 mg/L versus 1603 mg/L) is attributed to the lower cell density (7.35 versus 8.56 at 48 h) (Supplementary Fig. 6c and 6d). We added this data in the Result Section and gave a brief discussion in the Discussion Section.

5. Green or red fluorescent protein is used to analyze total cell populations. This method may not determine the live population ratio. Can they use CFU counting to further confirm the population dynamics?

Response: Thanks for the reviewer's comment. At the beginning of this work, we tested both the CFU-counting-based and the fluorescence-based methods. We observed that the colony number on a plate is not proportional to the dilution rate. Thus, the former is inaccurate and laborious. By contrast, the latter can count 100,000 single cells in a few seconds, although including both live and dead cells. Generally, the latter is a better method and can reflect the relative population dynamics of different coculture systems.

6. The authors may consider to add population kinetic models to explain and simulate the subpopulation interactions (e.g., Algal Research 58, 102372). The model can be helpful to explain how inoculation ratio and relative growth rates affect cell population dynamics.

Response: Thanks for the reviewer's suggestion. We agree that kinetic modeling is helpful to reveal growth dynamics and population interaction. The coculture system designed here are stable and insensitive to the inoculation ratios. Due to the lack of relative experience, we performed the metabolomic analysis instead, and discussed in the Discussion section the promising application of kinetic modeling in guiding the design and explaining the dynamics of the coculture systems.

7. The paper contains some typos and minor grammatical errors (e.g., Fig 4 missed a sub-fig caption g; line 409 miss a temperature symbol). The authors need to proofread the paper again.

Response: As suggested, the manuscript was proofread to correct the typos and grammatical errors.

Reviewer #2 (Remarks to the Author):

In this manuscript, authors engineered new microbial consortia based on multiple-metabolite cross-feeding, and a salidroside-producing coculture system was developed from it, with a yield of 12.52 g/L. Then, a caffeate-responsive biosensor was used in a coniferol-producing microbial consortia, which significantly increased the yield. Finally, a three-strain system was constructed for production of silybin and isosilybin. This study provides a new strategy for modular coculture engineering. The MS is suggested for publication in Nature Communications after the below issues are addressed.

1. In the introduction, the authors described “However, the coculture systems designed to date require adjustment of the initial inoculation ratios (IIRs) to optimize, severely limiting use for large-scale production”. The relevant references should be cited to support this.

Response: Thanks for the reviewer’s comment. The relevant references were cited to support this statement.

2. Still in the introduction, the authors described “Previous symbiotic relationships have been established primarily based on single-metabolite cross-feeding, with loose correlations between the strains that are prone to disruption”. It needs more concrete data from the previous reports to support this point.

Response: Thanks for the reviewer’s comment. We revised the statement and cited the relevant reference.

3. Again, in the introduction, the authors described “Self-regulation is another desired feature of a robust microbial coculture, where the strain ratio is autonomously coordinated to maximize metabolic flux of the product with minimal build-up of the intermediates.” Here, suggest to use a more conservative word to replace “maximize”, because it is difficult to say a population ration of the coculture strains established by biosensor (including the caffeate-responsive biosensor in this MS) is the best ration for compound production.

Response: Thanks for the reviewer’s suggestion. We replaced replace “maximize” to “increase”.

4. Please add more detailed description to define three coculture systems (neutralism, commensalism and mutualism).

Response: Thanks for the reviewer’s suggestion. More detailed description was added to define the three coculture systems (Lines 107, 141).

5. The abbreviation should be defined at the first place in the main text, for examples ptsG, manXYZ and glk. In this regard, it is strongly recommended to include more background of selected enzymes/genes in the metabolic engineering studies.

Response: Thanks for the reviewer's suggestion. More information about these genes was added (Line 98).

6. To accurately quantify the compounds, the standard curve of each compound should be plotted and must make sure the concentration of each sample for analysis is within the linear region.

Response: Thanks for the reviewer's suggestion. The standard curve of each compound was provided in the Supplementary Fig.12. The samples were properly diluted to make sure the accuracy of quantification.

7. In line 44, should reference 10 be reference 11?

Response: It should be reference 11. We corrected this error.

Reviewer #3 (Remarks to the Author):

In this manuscript, Li et al. reported a few new strategies to design microbial consortia. Their main objective was to develop consortia that are less sensitive to the initial inoculation ratio (being more robust). To this end, they constructed two strains that could use glucose and glycerol, respectively. They found that the glucose-utilizing strain (Bglc1) grew much faster than the glycerol strain (Bgly1) when being cocultured in a medium containing both glucose and glycerol. This led to Bglc1 dominating the final population. To balance the ratio of the two strains, they engineered Bglc1 so that it cannot synthesize glutamate from alpha-ketoglutarate using ammonium or glutamine as the amino donor (new strain: Bglc2); they modified Bgly1 so that it cannot synthesize any TCA intermediates (new strain: Bgly2). They confirmed that Bglc2 and Bgly2 could 1) exist in a balanced ratio when cocultured in the medium and 2) be used to synthesize salidroside efficiently. They further introduced biosensors into the coculture for improving synthesis of new target molecules. They also expanded the system to demonstrate that a three-member cross-feeding coculture could also be constructed using a similar concept.

I think the reported method is interesting and potentially better than existing ones. I wish to request the authors to address the following concerns to improve their manuscript:

1. Is it necessary to use such complicated design? The authors needed to introduce many cross-feeding designs to balance the two subpopulations. A reason is that Bglc1 grew much faster than Bgly1. When Zhang et al. reported their glucose/xylose coculture in 2015 (PNAS), it seems that they did not encounter a similar problem, possibly because their glucose-utilizing strain and xylose-utilizing strain had similar growth rate. It would be helpful if the authors could test salidroside production using the glucose/xylose strategy, which can be considered as a commonly used strategy in this field. The comparison would help assess the value of the new strategy being reported here. Bgly1 can utilize xylose but not glucose, so it can be used as the

xylose strain. It should be straightforward to disable xylose utilization to get a new glucose strain for the glucose/xylose coculture.

Response: As suggested, we tested salidroside production using the glucose/xylose strategy. The results showed that the mutualistic coculture is still better than the other two cocultures. Liu et. al reported salidroside production using the glucose/xylose coculture. In their study, the inoculation ratio has significant effect on salidroside titer (670 mg/L at 2:1; 130 mg/L at 1:2) (Metabolic Engineering, 2018, 47:243–253). By contrast, our design shows better stability (646 mg/L at 4:1 and 540 mg/L at 1:4). In this study, all the experiments were performed using the mixture of glucose and glycerol as the carbon source. To keep the fluency, these data were not included in the revised manuscript.

2. What are the metabolite exchanges Bgly2 had with Bglc2 when they are cocultured in the glucose/glycerol medium? The authors only mentioned that co-culturing with Bglc2 restored the growth of Bgly2 but did not specify the details of the interactions. A further investigation at the molecular level would substantially improve the depth of this study. There are many interesting questions that can be asked. For example, what are the key metabolites being transported from Bglc2 to Bgly2? Is it required to transport many TCA intermediates or only one critical metabolite? The authors may consider to answer the question through 1) ¹³C-Metabolic Flux Analysis, 2) Transcriptome analysis, and/or 3) Perturbing the interaction through gene deletion/inactivation.

Response: Thanks for the reviewer's suggestion. We performed the metabolomic analysis of the supernatants of the neutral and the mutualistic cocultures (Lines 169-178). A total of 5 TCA cycle intermediates and 9 amino acids were detected in the supernatants of both cocultures, among which the levels of α -ketoglutarate and glutamate were significantly higher in the mutualistic coculture than in the neutralistic coculture (Supplementary Fig. 4). Considering the essential roles of α -ketoglutarate and glutamate in the TCA cycle and amino acid metabolism, we infer that they may serve as the key metabolites for crossing feeding while the other relevant metabolites (carboxylic acids in the TCA cycle and amino acids) detected and even non-detected can also contribute to the cross-feeding.

Reviewer #4 (Remarks to the Author):

In order to create robust biomanufacturing-focused cocultures, the authors examine the effects of multi-metabolite cross-feeding on strain proportion and product titer. They begin by noting that the maintenance of cocultures via single-metabolite exchange is prone to variability and sensitive to the initial inoculation ratio (IIR). They propose to address this by increasing the codependence of each strain using multi-metabolite cross-feeding. First, the authors analyzed how the serial implementation of this strategy (moving from neutralism to commensalism and

then to mutualism) affected the final strain ratio given different IIRs. Then, they compared the same systems in their ability to efficiently synthesize salidroside after giving one half of the metabolic pathway to each constituent member. They also examined the effect of tuning these ratios by tying the growth of the less-dominant strain to the intermediate of an engineered coniferol pathway. Finally, they investigated the potential of this strategy to synthesize more complex molecules like silybin by comparing the biosynthetic efficiency of two- and three-strain implementations of this pathway.

The authors find that their chosen mutualistic relationship successfully holds their cocultures to tunable strain ratios which can then be used to dramatically increase biosynthetic efficiency. They first find that a wide range of IIRs do not disrupt the system's final strain ratio. They also find that maintaining and tuning this ratio via intermediate sensing improves their ability to synthesize salidroside and coniferol, again at a wide range of IIRs. Finally, they find that they are able to improve this efficiency even further in more complex pathways by splitting up the biosynthesis of silybin into three-strains while tuning the growth of one of the less-dominant strains. Based on its clear presentation and strong improvement over cited alternatives, I believe this paper could represent a strong contribution to this journal with a few important revisions.

Chiefly, the authors argue throughout the paper that their knockouts of the *ppc*, *gdhA*, and *gltBD* genes represent a definite example of the exchange of multiple metabolites between two strains. However, they demonstrate this by serially adding missing nutrients in monoculture and examining the recovered growth. I would expect similar growth improvements (between single and multi-metabolite growth) in non-deficient cultures, as one would expect in the comparison of growth in rich vs. minimal media. Without a direct experimental comparison to non-deficient cultures (and ideally individual knockout cultures as well), it is difficult to say that these strains are actually exchanging all of these metabolites. I would suggest either providing new evidence, as outlined above, or revising the claims made in this paper to make it clear that this multi-metabolite is a hypothetical explanation for the results seen in this paper.

Response: Thanks for the reviewer's suggestion. We performed the extracellular metabolomic analysis of the neutral (Bgly1 and Bglc1) and the mutualistic (Bgly2 and Bglc2) cocultures (Lines 169-178). A total of 5 TCA cycle intermediates and 9 amino acids were detected in the supernatants of both cocultures, among which the levels of α -ketoglutarate and glutamate were significantly higher in the mutualistic coculture than in the neutral coculture (Supplementary Fig. 4). Considering the essential roles of α -ketoglutarate and glutamate in the TCA cycle and amino acid metabolism, we infer that they may serve as the key metabolites for crossing feeding while other metabolites (carboxylic acids and amino acids) detected and even non-detected can also contribute to the cross-feeding.

I would also like to see the following addressed:

In general, the conceptual metabolic pathway schematics are difficult to read. Figure 5a has so many overlapping lines in the center that it is almost impossible to interpret.

-Some more depth on the statistical analysis of this data would be appreciated.

Response: Thanks for the reviewer's suggestion. We revised the schematics. The biosynthetic pathways of salidroside, coniferol, taxifolin and silybin/isosilybin were combined into one detailed map (Supplementary Fig. 5) while the simplified pathways were shown in Figures 3, 4 and 5.

The data analysis was carried out in more depth.

Besides their different biosynthetic pathways, what is the relationship between the two glucose-dependent strains of section 2.4? Are they missing the same nutrients? If they contain the same knockouts (and therefore require the same nutrients from the glycerol-dependent strain), this could possibly further undermine the hypothesis that multi-metabolite exchange is responsible for the improved efficiency reported here. This is especially concerning since the authors did not vary IIR in this final section.

Response: Thanks for the reviewer's comment. The two glucose-dependent strains have the same genetic background. We think that the improved efficiency of silybin/isosilybin production by the three-strain coculture is due to (1) the alleviated metabolic burden of the host cells and (2) the balanced supply of the two direct precursors (coniferol and taxifolin). As shown in Fig. 5d, the downstream strain Bglc2-Sil only takes around 40 % in the total population, leading to the accumulation of large amount of caffeate. In the three-strain coculture, the total population ratio of the downstream strains Bglc2-Con(b) and Bglc2-Sil(b) reaches 80 %. In addition, compared with coniferol, the synthesis of taxifolin is rate-limiting. Therefore, Bglc2-Sil(b) is equipped with the caffeate-sensor to increase its population ratio and consequently the supply of taxifolin.

Reviewers' Comments:

Reviewer #1:

Remarks to the Author:

My concerns have been addressed.

Reviewer #2:

Remarks to the Author:

The concerns have been well addressed in the revised manuscript. It is suggested to accept the MS for publication.

Reviewer #3:

Remarks to the Author:

I appreciate the new experiments the authors did to address the two concerns I have raised. Below are my replies:

1. It is okay not to include the new glucose/xylose co-culture data in the manuscript, but I request to review the technical details of this experiment to see if a fair comparison has been done. The authors should at least describe the used strains and culture conditions.

2. The measured metabolite pool size data are not informative. Their change does not reflect changes to the related carbon flux. I would suggest the authors to consider one of the three experiments I suggested to improve the depth of this study.

Reviewer #4:

Remarks to the Author:

I believe the authors have adequately addressed most of the reviewers' comments.

Response to the referees

REVIEWER COMMENTS

Reviewer #1 (Remarks to the Author):

My concerns have been addressed.

Thanks the reviewer's efforts in improving the quality of the manuscript.

Reviewer #2 (Remarks to the Author):

The concerns have been well addressed in the revised manuscript. It is suggested to accept the MS for publication.

Thanks the reviewer's efforts in improving the quality of the manuscript.

Reviewer #3 (Remarks to the Author):

I appreciate the new experiments the authors did to address the two concerns I have raised.

Below are my replies:

1. It is okay not to include the new glucose/xylose co-culture data in the manuscript, but I request to review the technical details of this experiment to see if a fair comparison has been done. The authors should at least describe the used strains and culture conditions.

Response: Thanks for the reviewer's suggestion. The technical details of the experiment are provided as follows.

Strains, culture media and conditions

The strains used for the glucose/xylose coculture are listed in Table S1. Strains Bxyl1-Tyr/Bglc3-Sal were used for the Neutralism co-culture, strains Bxyl1-Tyr/Bglc4-Sal were used for Commensalism co-culture, and strains Bxyl2- Tyr/Bglc4-Sal were used for Mutualism co-culture.

Luria-Bertani (LB) medium containing 10 g/L tryptone, 10 g/L NaCl and 5 g/L yeast extract was used for seed propagation. The strains were cultivated in the M9 media containing 6.78 g/L Na₂HPO₄, 3 g/L KH₂PO₄, 1 g/L NH₄Cl, 0.5 g/L NaCl, 1 mM MgSO₄, 0.1 mM CaCl₂, 15 g/L glucose and 15 g/L xylose. Trace elements were also supplemented to all batch fermentations, giving final concentrations of 0.03 mg/L H₃BO₃, 0.38 mg/L CuCl₂, 0.4 mg/L Na₂EDTA, 0.5 mg/L CoCl₂, 0.94 mg/L ZnCl₂, 1.6 mg/L MnCl₂, 3.6 mg/L FeCl₂, and 100 mg/L thiamine.

Ampicillin, kanamycin, and chloramphenicol were added to the media when needed at final concentrations of 100, 50, and 34 mg/mL, respectively.

Shake flask experiments were performed at 37 °C and 220 rpm. Overnight seed cultures were diluted 1:50 into 50 mL fresh M9 media, and induced with 0.5 mM isopropyl-β-D-thiogalactoside (IPTG) at 3 h after inoculation. Samples were taken every 12 hours. Cell growth was monitored by measuring the optical density at 600 nm (OD600) and product accumulation were analyzed with high-performance liquid chromatography (HPLC).

Table S1 Strains utilized in the glucose/xylose co-culture experiment

Strains	Description	Source
Bxyl1	BW25113Δ pykA Δ pykF Δ ptsG Δ glk Δ manXYZ	This study
Bglc3	BW25113Δ xylA	This study
Bxyl2	BW25113Δ pykA Δ pykF Δ ptsG Δ glk Δ manXYZ Δ ppc	This study
Bglc4	BW25113Δ xylA Δ gdhA Δ gltBD	This study
Bxyl1-Tyr	Bxyl1, pZE12-luc and pCS-TPTA and pSA-mcherry	This study
Bglc3-Sal	Bglc3, pZE-ugt85A1 and pCS-pg and pSA-AA	This study
Bxyl2-Tyr	Bxyl2, pZE12-luc and pCS-TPTA and pSA-mcherry	This study
Bglc4-Sal	Bglc4, pZE-ugt85A1 and pCS-pg and pSA-AA	This study

Note: The strains Bgly1, Bgly2, Bgly1-Tyr and Bgly2-Tyr were renamed to Bxyl1, Bxyl2, Bxyl1-Tyr and Bxyl2-Tyr, and used for the glucose/xylose coculture experiment.

Results

As shown in Fig. S1, in all the three cocultures the percentages of the upstream xylose-utilizing strains decreased rapidly in the first 12 h. After 12 h, the percentage of strain Bxyl1-Tyr remained low, resulting in a low final salidroside titer (159 mg/L, Fig S2a). On the contrary, the percentages of strain Bxyl2-Tyr in both the commensalistic and mutualistic cocultures kept increasing after 12 h, and the mutualistic coculture still showed better stability than the commensalistic coculture in both the population composition (Fig. S1 b and c) and salidroside production (Fig. S2 b and c).

The superiority of the commensalistic coculture over the mutualistic coculture in the glucose/xylose media is not as significant as that in the glucose/glycerol media. This should be due to the slower growth of the strains in glucose/xylose media, which also leads to drastic decrease in salidroside production.

Fig. S1 Curves of cell growth and population proportion change in the glucose/xylose co-culture experiment. **a** the Neutralism Bxy11-Tyr/Bglc3-Sal coculture, **b** the Commensalism Bxy11-Tyr/Bglc4-Sal coculture, **c** the Mutualism Bxy12-Tyr/Bglc4-Sal coculture. Data shown are mean \pm SD (n=3 independent experiments).

Fig. S2 Comparison of salidroside titers in the three coculture systems using xylose/glucose as substrates. Data shown are mean \pm SD (n=3 independent experiments).

2. The measured metabolite pool size data are not informative. Their change does not reflect changes to the related carbon flux. I would suggest the authors to consider one of the three experiments I suggested to improve the depth of this study.

Thanks for the reviewer's suggestion. We agree that it is a good direction to investigate the detailed interactions between the coculture members, which is a common interest and also a challenge for researchers in this field. ^{13}C metabolic flux analysis (^{13}C -MFA) is a useful experimental approach to elucidate detailed metabolic fluxes in biological systems. According to the literature, ^{13}C -MFA has been applied almost exclusively to mono-culture systems. Recently, the Antoniewicz group developed a novel approach for performing ^{13}C -MFA of co-culture systems and demonstrated for the first time the possibility to estimate metabolic flux distributions in multiple species simultaneously (Metabolic Engineering 2015, 31:132-139). ^{13}C -MFA is a systematic project. Due to the lack of relevant infrastructures and experience, it

may take years for us to complete this experiment. We would like to conduct collaborative research in this topic in the future.

For the transcriptome analysis, it is difficult to distinguish the coculture members, especially strains from the same species. Gene deletion/inactivation may disturb and even paralyze the coculture system, but may not provide direct evidence of the interactions.

Generally, our current data is sufficient and can support our conclusions. We hope the reviewer agrees with us and that our research can be shared with scientific community.

Reviewer #4 (Remarks to the Author):

I believe the authors have adequately addressed most of the reviewers' comments.

Thanks the reviewer's efforts in improving the quality of the manuscript.

Reviewers' Comments:

Reviewer #3:

Remarks to the Author:

The authors have addressed all my concerns.